# Selecting Thresholds of Heat-Warning Systems with Substantial Enhancement of Essential Population Health Outcomes for Facilitating Implementation

**DOI:** 10.3390/ijerph18189506

**Published:** 2021-09-09

**Authors:** Shih-Chun Candice Lung, Jou-Chen Joy Yeh, Jing-Shiang Hwang

**Affiliations:** 1Research Center for Environmental Changes, Academia Sinica, Taipei 11529, Taiwan; joyyeh100@gate.sinica.edu.tw; 2Department of Atmospheric Sciences, National Taiwan University, Taipei 10617, Taiwan; 3Institute of Environmental and Occupational Health Sciences, National Taiwan University, Taipei 10617, Taiwan; 4Institute of Statistical Science, Academia Sinica, Taipei 11529, Taiwan; hwang@sinica.edu.tw

**Keywords:** health adaptation, Sustainable Development Goal 3, heat-health threshold identification, extreme events and health

## Abstract

Most heat-health studies identified thresholds just outside human comfort zones, which are often too low to be used in heat-warning systems for reducing climate-related health risks. We refined a generalized additive model for selecting thresholds with substantial health risk enhancement, based on Taiwan population records of 2000–2017, considering lag effects and different spatial scales. Reference-adjusted risk ratio (RaRR) is proposed, defined as the ratio between the relative risk of an essential health outcome for a threshold candidate against that for a reference; the threshold with the highest RaRR is potentially the optimal one. It was found that the wet-bulb globe temperature (WBGT) is a more sensitive heat-health indicator than temperature. At lag 0, the highest RaRR (1.66) with WBGT occurred in emergency visits of children, while that in hospital visits occurred for the working-age group (1.19), presumably due to high exposure while engaging in outdoor activities. For most sex, age, and sub-region categories, the RaRRs of emergency visits were higher than those of hospital visits and all-cause mortality; thus, emergency visits should be employed (if available) to select heat-warning thresholds. This work demonstrates the applicability of this method to facilitate the establishment of heat-warning systems at city or country scales by authorities worldwide.

## 1. Introduction

In facing the climate emergency [1], adaptation is as important as mitigation. Record-breaking temperatures with longer durations have been occurring more frequently worldwide [2,3,4]; the resulting high casualty highlights the urgency and importance of implementing health adaptation strategies to meet the third Sustainable Development Goal (SDG) set by the United Nations [5] for health risk reduction. Accompanied by countermeasures, a heat warning system can be viewed as a critical health adaptation strategy needed in all countries [6], which could trigger adequate responsive actions by the authorities and advise people for self-protection to prevent high casualty levels. Particularly, in tropical and subtropical countries where inhabitants deem themselves already adapted to a hot climate—which is not true, as has been shown in various studies [7,8,9,10]—heat-related health risks under climate change are given insufficient attention and heat warnings may even be ignored. While heat warning systems have long been recognized as the most effective way to reduce heat-health risks [11,12], selecting a proper heat indicator and an appropriate threshold with sound scientific evidence remain challenging [7,13]. Scientific tools are needed to facilitate the implementation of heat warning systems.

The effectiveness of heat warning systems has been evaluated by Toloo et al. [14,15], for instance. They conducted a systematic review of 15 studies and found that the number of expected deaths were reduced after the implementation of heat warning systems accompanied with responsive actions, based on six articles acquired in 2013. Nevertheless, the effectiveness of such systems in reducing morbidity remained an open question and required more studies. In responding to climate change, more countries have established heat warning systems in recent years. Casanueva et al. [16] reviewed the existing heat warning systems in 16 European countries in 2019 and found that they are quite diverse and use different indicators with thresholds defined differently. The majority of those systems are based on temperature only, while in Germany and Austria, they are based on a thermal-stress index considering temperature, radiation, wind, and humidity. The thresholds were defined mostly based on epidemiological (mortality) evidence with others based on climatological percentiles. Additionally, since outdoor workers are especially vulnerable towards heat, the pan-European HEAT SHIELD project has developed an occupational heat-health warning system, in order to reduce the heat-health impacts of outdoor workers [17]. This warning system uses the wet-bulb globe temperature (WBGT) as a heat indicator, which is described later. For any chosen heat indicator, a systematic method to select proper thresholds is the foundation for an effective heat-health warning system. The current work intends to provide such a systematic method with epidemiological models.

It has been recognized that co-creating knowledge is essential for translating scientific evidence into actions for sustainability [18,19]. In our case, the scientific method was revised taking into account policy concerns after interacting with policy makers. We discussed with the Central Weather Bureau (CWB), which is responsible for establishing a heat warning system in Taiwan, in order to understand their concerns, in several stakeholder engagement meetings over the years. The CWB officials agreed that the determination of a proper threshold for issuing heat warnings should ideally be supported with historical records of “significant” heat-health impacts. However, traditional epidemiological studies have failed to provide such thresholds. As emphasized in our earlier work [7], most heat-health studies have, in fact, assessed the starting points in the right arm of the U-shaped or V-shaped temperature-related health impacts, just outside the comfort zone of human being, which is associated with statistically (but not substantially) increased health impacts. These starting points are often too low to be used in heat-warning systems. For example, if the heat-health warning threshold in Taipei was chosen based on the thresholds identified in typical epidemiological studies shown in publications focusing on the starting points of mortality increase, heat warnings would be announced 134 days in summer of 2017, i.e., almost the entire summer [7]. Warnings that are too frequently issued may cause the general public to ignore them, thus rendering them ineffective. Thus, the thresholds of the heat-related mortality or morbidity increase identified by typical epidemiological studies are often not appropriate thresholds for issuing a heat warning in practice. Especially for the developing countries, the required manpower and resources associated with governmental intervention programs is also a limiting factor for a prolonged heat-health warning.

These “statistically significant” thresholds may be important to scientists, but not practical for use in the real-world situations, as summer-long alerts would be issued based on these starting points. The CWB officials were concerned that such frequent warnings would fail to fully capture the attention of the general public while exhausting the resources of the authorities. From the viewpoint of policy makers, issuing heat warnings should be associated with “substantial health impacts”, and the intervention measures thus triggered should serve to reduce significant health risks. After realizing the policy concerns, a multidisciplinary collaboration was formed with environmental health scientists and statisticians, in order to refine current statistical methods and identify the heat-health thresholds with “statistically significant” and “substantial health impacts”, thus meeting both the criteria of scientists and policy makers.

To date, only a few studies have focused on providing a systematic approach to assist governmental agencies in establishing proper heat-warning thresholds. With data from the USA, Petitti et al. [20] attempted to identify multiple trigger points at which heat-health intervention measures might be activated, using the 95% confidence interval as a criterion for selecting increasing risk temperature and excess risk temperature above the minimum risk temperature. Our earlier work proposed a more straightforward approach for simultaneously evaluating proper thresholds with substantial health risk increments according to local health records, thus providing a systematic method for selecting proper thresholds [7]. On the basis of our previous work, this study further modifies the method to selecting proper thresholds, not only with emphasis on substantial health risk enhancement, but also emphasizing both lag effects (0–2 days) and with at-risk populations of different sizes taken into account. The flexibility of applying this useful scientific tool at different spatial resolutions to assist authorities in establishing a heat warning system with solid health-based evidence is demonstrated.

As evaluated in our previous work [7], WBGT, compared against both temperature and apparent temperature, showed the most obvious increasing trends for three different health outcomes in the Taiwanese population, making it a proper heat-health indicator. Moreover, WBGT comprises four essential meteorological parameters related to heat stress; namely, temperature, relative humidity (RH%), wind speed, and solar radiation. A proper heat-warning indicator should be related to human physiological changes with extensive physiological-based evidence [21], rather than considering only subjective thermal comfort such as apparent temperature. In addition, the WBGT has been used for six decades as an indicator to prevent heat stress-related health impacts in workplaces; occupational health studies have demonstrated the relationships between WBGT and heat-related health outcomes of workers [21,22,23,24,25,26]. Increasing evidence has also shown that WBGT is a suitable heat-health indicator for the general public [10,27,28,29,30]; however, the warning thresholds for workers cannot be directly applied to the general public including vulnerable populations, such as the elderly and children. Thus, this work aims to identify proper WBGT thresholds for the general public.

In summary, to fill the scientific gap of a feasible health evidence-based method for the authorities to identify proper thresholds for heat warning systems, the objectives of this work are as follows: First, a refined statistical method is proposed for identifying proper thresholds, taking into consideration one- to three-day substantial health risk enhancement, with data from Taiwan used for illustration. Second, proper thresholds are selected using different health records (heat-related emergency and hospital visits, as well as all-cause mortality) and with both WBGT and temperature for comparison. Third, the applicability of the proposed method is examined at different spatial scales with at-risk populations of different sizes and heat events of different frequencies. The methods and findings presented in this study should provide useful reference and assistance to central and local governments worldwide in selecting proper heat-warning thresholds, according to their own health records. Establishing such an evidence-based heat-health warning systems in cities and regions in different climate zones across the globe will speed up the progress toward the No. 3 of SDG (SDG3, ensure healthy lives and promote well-being for all at all ages) with reduced heat-health risks.

## 2. Materials and Methods

As discussed in our earlier publication [7], most heat-health studies identified thresholds just outside human comfort zones, which are often too low to be used in heat-warning systems. Due to the aforementioned reasons, we revised the typical epidemiological models and tried to identify thresholds with “statistically significant” and “substantial health impacts” which can be used to establish an effective heat warning system in order to reduce health risks. The following describes the data and revised models used for this purpose.

### 2.1. Health Records

Records on daily emergency and hospital visits of heat-related illness between 2000 and 2017 were obtained from the database of the Health and Welfare Data Science Center of the Ministry of Health and Welfare. This database contains hospital visit information of almost the entire population of Taiwan (23.71 million), including the age and sex of patients, as well as hospital location. Cases of heat-related illness (according to the 9th Revision of the International Classification of Diseases, ICD9: 992, including heat stroke and heat exhaustion, for 2000–2015 data, and ICD10: T67.0XXA–T67.9XXA for 2016–2017 data) were used in this work. All-cause mortality counts (excluding accidents and suicide) between 2008 and 2017 were obtained from the Taiwan National Mortality Registry. Data before 2008 were not used, as the location of death was not specified. This study was reviewed and approved by the Institution Review Board of Academia Sinica, IRB No. AS-IRB-BM-18030. All health records were applied for use right after IRB approval and data after 2017 was not available when data analysis was carried out. Moreover, the format and variable definitions of the aforementioned databases were revised over the years. Records of daily emergency and hospital visits of 2000–2017 were comparable to be included in the same analysis; the same applied to mortality records of 2008–2017. The SAS 9.4 (SAS Institute Inc., Cary, NC, USA) and R 3.5.1 (with mgcv package) software were used.

### 2.2. Heat Indicator and Air Pollutant Data

Hourly meteorological data from 2000 to 2017 were obtained from 20 non-mountainous (<500 m above sea level) CWB stations in Taiwan. These CWB stations located at city or county centers near where residents live. Daily maximum temperatures of these stations were averaged for analysis in Taiwan as a whole. To evaluate the spatial difference in heat-health relationships and to assess the flexibility of the present method at different spatial scales, Taiwan island was divided into sub-regions; namely, North, Central, South, and East Taiwan (Appendix A
Appendix A). Geographically separated by mountain ranges, North, Central, and South Taiwan are mainly on the western side of the island, with much higher population density compared to the eastern side. The populations and areas of these sub-regions in 2017 are listed in Table 1. There are six, two, six, and six weather stations located in North, Central, South, and East Taiwan, respectively.

WBGT is a weighted combination of dry-bulb temperature, natural wet-bulb temperature, and globe temperature [21]. However, the natural wet-bulb temperature and globe temperature were not provided by routine meteorological measurements. Nevertheless, equations for calculating the WBGT, according to fundamental principles of heat and mass transfer, using standard meteorological data have been derived and validated [31]. Thus, with the hourly temperature, RH%, wind speed, and solar radiation from CWB stations as input, the WBGT can be calculated. The daily maximum WBGTs of these stations were also averaged for analysis.

In the heat-health model, confounders such as air pollutants [32] were adjusted, using daily means of PM_2.5_ levels from 2000 to 2017; PM_2.5_ was used, as it is currently the most concerning pollutant in Taiwan, and it has high correlation with other pollutants. PM_2.5_ levels were obtained from 58 ambient air-quality monitoring stations and four industrial stations located in populated areas (population density close to or above median population density of Taiwan in 2017, 618 people/km^2^; Table 1) of the Environmental Protection Administration, Taiwan, between 2000 and 2017. For North, Central, South, and East Taiwan, 25, 11, 22, and four stations, respectively, are included in the present analysis. In East Taiwan, no PM_2.5_ data were available for 2000–2004; thus, the heat-health model could only run for 2005–2017.

### 2.3. Models for Threshold Selection

The models used were based on our previous work [7], with three modifications considering (1) risk comparisons with ratios, (2) lag effects, and (3) rare event interference, as described in the following paragraphs. Briefly, we first confirmed the linear patterns of health outcomes above certain cut-points of the heat indicators, then evaluated the relative risks (RRs) of the different threshold candidates and compared those RRs with a reference. The best threshold is chosen on the basis of RR comparison with the occurring frequency of alerts examined.

Only data from May to October (warm season) were used for modeling, thus eliminating the need to consider seasonal effects. Data of health outcomes were fitted to generalized additive models (GAMs) with Poisson distribution, in order to examine the relationships between heat indicators (denoted as $D_{t}$, WBGT or temperature) and health outcomes (denoted by $Y_{t}$, heat-related emergency visits, heat-related hospital visits, and all-cause mortality counts). The initial model was specified as:(1)$$\mathrm{Log}\left[E\left(Y_{t}\right)\right]=\alpha +f\left(D_{t}\right)+g\left(t\right)+\sum_{i=1}^{m}\lambda_{i}C_{it},$$
where α is a constant, $f\left(D_{t}\right)$ is a thin plate spline function of the heat indicator with a number of knots over the days in each warm season, and $g\left(t\right)$ is another thin plate spline function for day *t,* with some knots for examining the daily change pattern during the warm seasons. Potential covariates on day *t*, denoted by $C_{it}$, such as daily mean PM_2.5_ concentration, day of week, holiday, and periods prevalent with the severe acute respiratory syndrome, were considered for adjustment. The number of knots for $f\left(D_{t}\right)$ was automatically chosen for different models. The number of knots h=3 was adopted for $g\left(t\right)$, according to lower Akaike information criterion values for each warm season; hence, $g\left(t\right)$ with h=54 and 30 was considered for modeling the 18-year data on heat-related emergency and hospital visits and 10-year all-cause mortalities, respectively. After obtaining the best model, the estimated smooth function $\hat{f}\left(D_{t}\right)$ was plotted, in order to examine the heat-health relationships. This revealed a linear heat-health relationship for the heat indicator beyond certain cut-points. Then, this model was further modified to determine proper thresholds. Specifically, the daily maximum heat indicator on day *t* in the model was replaced by $H_{t}$, defined as:$$H_{t}=\left\{\begin{array}{c} D_{t}-\theta ,\;\mathrm{if}\;D_{t}>\theta \\ 0,\;\mathrm{otherwise} \end{array}\right.,$$
where θ is a specified threshold candidate of the heat indicator. The model is then modified as:(2)$$\mathrm{Log}\left[E\left(Y_{t}|\theta \right)\right]=\alpha +\beta_{\theta }H_{t}+g\left(t\right)+\sum_{i=1}^{m}\lambda_{i}C_{it},$$
where $\beta_{\theta }$ is the damage coefficient associated with a unit increase of the heat indicator from the chosen threshold θ. The relative risk (RR), $e^{k\times \beta_{\theta }}=E\left(Y_{t}|\theta ,D_{t}=\theta +k\right)/E(Y_{t}|\theta ,D_{t}=b_{\theta }$), for any value $b_{\theta }\leq \theta$, is defined by the ratio between the expected health outcome under exposure to a heat indicator value of *k* units over the threshold candidate and the expected health outcome for any exposure level below or equal the threshold candidate. The threshold candidates were assessed starting from 30 °C (the lowest threshold candidate) at 0.5 °C increments sequentially, in view of higher heat-related emergency and hospital visits in Taiwan at WBGT and temperature above 30 °C [7]. The models were repeatedly fitted with the threshold candidates until the sample size was too small (i.e., very few cases or days fell above the threshold candidate).

The RR values of these threshold candidates were compared with that of a reference, the chosen lowest threshold candidate (denoted by a=30 °C in this work). The RR difference, calculated as $\left[e^{\beta_{\theta }}-e^{\beta_{a}}\right]\times 100\%$, was used in our previous work [7]. We further modified the comparison of these RRs, expressed as a ratio rather than a difference, to emphasize the enhancement of the expected health outcome above threshold candidates. This further-refined indicator, the reference-adjusted risk ratio (RaRR), is defined as:(3)$$RaRR=\frac{e^{k\times \beta_{\theta }}}{e^{k\times \beta_{a}}}=\frac{E\left(Y|a,D_{t}=b_{a}\leq a\right)}{E\left(Y|\theta ,D_{t}=b_{\theta }\leq \theta \right)}\times \frac{E\left(Y|\theta ,D_{t}=\theta +k\right)}{E\left(Y|a,D_{t}=a+k\right)}\;\mathrm{for}\;\theta \geq a.$$

Note that the definition of RR indicates that $b_{\theta }$ could be any value smaller than or equal to θ. When we set $b_{\theta }=b_{a}=a\leq \theta$, the first factor of the multiplication in Equation (3) would be close to 1, and the RaRR would approximate $E\left(Y|\theta ,D_{t}=\theta +k\right)/E\left(Y|a,D_{t}=a+k\right)$, which could be interpreted as the relative risk of the expected health outcome for exposure to the level of *k* units above the heat indicator candidate, compared with that for exposure to same *k* units above the reference. We determine the threshold as the candidate θ with the maximum RaRR, for which the corresponding model estimate ${\hat{\beta }}_{\theta }$ is statistically significant. The selected threshold often indicates that the expected health outcome of the population exposed to WBGT or temperature above the threshold has the maximum increase among those heat indicator values above the reference.

When the two-day lag effect is considered, the equation is modified as:(4)$$\log\left[E\left(Y_{t}|\theta \right)\right]=\;\alpha +\beta_{\theta 0}H_{t}+\beta_{\theta 1}H_{t-1}+\beta_{\theta 2}H_{t-2}+g\left(t\right)+\sum_{i=1}^{m}\lambda_{i}C_{\mathrm{it}},$$
where $H_{t}=\left\{\begin{array}{c} D_{t}-\theta ,\;if\;D_{t}>\theta \\ 0,\;\mathrm{otherwise} \end{array}\right.$. The same definition is applied for $H_{t-1},\;H_{t-2}.$

Moreover, in order to evaluate spatial differences in the heat-health relationship and the flexibility of this model with at-risk populations of different sizes, the above models were applied to the sub-regions of Taiwan. However, above certain heat indicator values, the heat-health relationships became unstable. As can be seen in Figure 1a, there exists a linear relationship of WBGT with heat-related hospital visits for the whole of Taiwan island, but the number of heat-related hospital visits in Central Taiwan fluctuated when the WBGT exceeded 33.0 °C (Figure 1b). These fluctuations may be attributed to the infrequent occurrence of days above 33.0 °C (rare events) in Central Taiwan. To avoid the interference of such rare events, the upper limit δ chosen for the modified model approximated the 99.5th percentile of the heat indicators in the specific sub-region (Table 1), as expressed in Equation (5):(5)$$\log\left[E\left(Y_{t}|\theta \right)\right]=\alpha +\beta_{\theta 0}H_{t}+\beta_{\theta 1}H_{t-1}+\beta_{\theta 2}H_{t-2}+\gamma_{0}L_{t}+\gamma_{1}L_{t-1}+\gamma_{2}L_{t-2}+g\left(t\right)+\sum_{i=1}^{m}\lambda_{i}C_{\mathrm{it}},$$
where $H_{t}=\left\{\begin{array}{c} D_{t}-\theta ,\;\mathrm{if}\;\delta >D_{t}>\theta \\ 0,\;\mathrm{otherwise} \end{array}\right.$ (the same definition is applied for $H_{t-1},\;H_{t-2})$ and $L_{t}=\left\{\begin{array}{c} 1,\;\mathrm{if}\;D_{t}\geq \delta >\theta \\ 0,\;\mathrm{otherwise} \end{array}\right.$ (the same definition is applied for $L_{t-1},\;L_{t-2}).$

For consistency, Equation (5) was applied for all spatial scales in this study. It should be noted that θ sometimes differs from δ by 0.5 °C in these models. Thus, the new RR between the expected health outcome used a unit of 0.5 °C increase (instead of the commonly used 1 °C increase) over the threshold candidate for identifying the threshold. For coherent comparisons among different regions, this new RR is used in the subsequent analysis, regardless of the actual differences between δ and θ at different spatial scales.

In summary, the present method first confirms the linear patterns of health outcome above certain cut-points of the heat indicators using Equation (1). Then, Equation (5) is used to estimate the new RR values for a 0.5 °C increase above each threshold candidate θ and a fixed reference $\left(a=30\;{}^{\circ}C\;\mathrm{for}\;\mathrm{this}\;\mathrm{study}\right),$ denoted as $e^{\beta_{\theta j}/2}$ and $e^{\beta_{aj}/2}$ for j=0, 1, 2. The threshold candidate θ with the highest $RaRR_{j}=e^{(\beta_{\theta j}-\beta_{aj})/2}$ for j=0, 1, 2 is potentially the optimal one. Afterwards, the historical occurring frequency above the selected thresholds is examined.

## 3. Results

### 3.1. Health Risks Associated with Increased WBGT and Temperature

Table 2 shows the summary of health and environment data in Taiwan during the warm seasons between 2000 and 2017. For the whole Taiwan island, daily maximum heat-related emergency and hospital visits reached 73 and 1533, respectively; and daily maximum all-cause mortality was 510 (Table 2a). Males had more heat-related emergency visits and higher daily all-cause mortality, while females had more heat-related hospital visits. Among different age groups, people aged 15–64 accounted for the majority of heat-related emergency and hospital visits, while those aged ≥65 had the highest counts in all-cause mortality. Among sub-regions, North Taiwan had the highest counts, while East Taiwan had the lowest counts in all three health outcomes. As for environmental data, the area-averaged daily maximum WBGT and temperature during the warm seasons of these 18 years for the whole Taiwan island were 33.6 °C and 34.7 °C, respectively (Table 2b). North and Central Taiwan had higher WBGT and temperature maxima than South Taiwan. This phenomenon may be counter-intuitive at first sight but was attributable to heat accumulation by the basin locations of the largest cities in North and Central Taiwan.

Table 3a,b shows RRs estimated by the modified GAMs for the whole Taiwan island, with different threshold candidates of WBGT and temperature, respectively, considering the lag effects of 0–2 days. For heat-related emergency and hospital visits with lag effects of 0, 1, and 2 days, RRs showed rising trends, with an increase in the WBGT candidate from 30 °C to 32.5 °C and temperature candidate from 30 °C to 33.5 °C. Moreover, the highest RRs with statistical significance among different lag days of both WBGT and temperature all occurred at lag 0 for heat-related emergency and hospital visits. For example, at a WBGT threshold of 32.5 °C, the highest RRs of heat-related emergency visits were 1.83 at lag 0. In contrast, RRs for all-cause mortality did not differ much among different lag days and threshold candidates for both WBGT and temperature.

### 3.2. RaRR Comparison among Different Categories

The novelty of this method is the proposed RaRR, which emphasizes the enhancement of RR against that of a reference. The RR values of different health outcomes at different WBGT threshold candidates and RaRR at the highest threshold candidate are presented in Appendix A (sex/age and sub-region, respectively), while those with temperature are displayed in Appendix A. Figure 2a,b show changes in the RaRR of heat-related emergency visits at different WBGT threshold candidates for Taiwan at the country level and in different sub-regions, as well as for different sex and age groups, respectively. The levels of statistical significance associated with the corresponding RR are denoted by different symbols. Equivalent information for heat-related hospital visits and all-cause mortality are illustrated in Figure 2c–f, respectively. Likewise, Figure 3a–f displays changes in RaRR at different temperature threshold candidates for the three different health outcomes under different categories.

For emergency and hospital visits of both WBGT and temperatures, the highest RaRRs among different lag days in the categories of sex, age, and sub-region mostly occurred at lag 0 (Figure 2a–d and Figure 3a–d). In the case of Taiwan with WBGT, the highest RaRR of heat-related emergency visits were seen at lag 0 (first panel of Figure 2a, RaRR = 1.44, Appendix A); while that of heat-related hospital visits occurred at lag 0 with a lower value (first panel of Figure 2c, RaRR = 1.19, Appendix A). On the contrary, the highest RaRR of all-cause mortality for the whole of Taiwan occurred at lag 1 (first panel of Figure 2e, RaRR = 1.05, Appendix A).

Variations in RaRR among different sub-regions at both WBGT and temperature thresholds (Figure 2a,c,e and Figure 3a,c,e) were observed. For heat-related emergency and hospital visits, the patterns of increasing RaRR along with increased WBGT and temperature were observed for most cases of lags 0–2 at all sub-regions, except for East Taiwan which had different patterns, likely due to the small-sized at-risk populations and small sample sizes for modelling (with data from only 2005–2017). For all-cause mortality, fewer RR were statistically significant in the sub-regions with all lag days for both WBGT and temperature.

In comparison, WBGT had higher RaRRs than temperature in most categories; and RaRRs of the heat-related emergency visits were all higher than the respective RaRRs of the heat-related hospital visits and all-cause mortality in the same category. Thus, the WBGT candidate with the highest RaRR of heat-related emergency visits were potentially the appropriate thresholds for the heat-warning system. The potential regional-specific thresholds were 32.5 °C, 34 °C, 32.5 °C, 32.5 °C, and 32 °C for the entire, North, Central, South, and East Taiwan, respectively. The following results and discussion are mostly focused on WBGT.

To protect vulnerable populations is the mission of the heat warning system. We assessed heat risks for different sex and age groups. The present findings revealed higher RaRRs of emergency visits for females at lags of 0–2 at WBGT of 32.5 °C than for males; for example, higher RaRR at lag 0 was observed in females (1.69) than in males (1.36). In contrast, RaRRs of hospital visits at WBGT of 32.5 °C at lag 0 were higher in males (1.22) than in females (1.18). For all-cause mortality with WBGT, the highest RaRRs for males at 32.5 °C occurred at lag 1 (1.05); while the RaRR at the same lag day for females was 1.04.

For different age groups, we found that, at a WBGT of 32.5 °C, the children had the highest RaRR of heat-related emergency visits at both lags of 0 (1.66) and 1 (1.81), followed by the elderly (Figure 2b, Appendix A); while the elderly was the only age group with statistically significant RRs at lag 2 (RaRR =1.26, Appendix A). Regarding hospital visits at a WBGT of 32.5 °C, the highest RaRR among the three age groups occurred at the age of 15–64 at lags 0 and 2 (1.19 and 1.15, respectively; Figure 2d, Appendix A). The 0–14 age group had the highest RaRR (1.25) at lag 1, followed by the elderly (1.21). For all-cause mortality at WBGT of 32.5 °C (Figure 2f), the elderly was the only age group with statistically significant RRs at lags of 0 and 1, with RaRR of 1.03 and 1.05 (Appendix A), respectively. At lag 2, the 15–64 age group was the only one with statistically significant RR, with an RaRR of 1.05 (Appendix A).

### 3.3. Potential WBGT Threshold Candidates

The alerts based on these thresholds would not be announced every day for the whole summer, as it would be based on the minimal health risks assessed in traditional epidemiological studies, as shown in Figure 4, with occurring frequency exceeding the specified WBGT thresholds classified by five categories for a ten-year period (2008–2017). One count in the figure represents one period of consecutive days, not one day, that met the specified criteria. Compared to applying one threshold for the entirety of Taiwan (32.5 °C), the occurring frequency of periods above the threshold were lower in the North and higher in the East with their respective thresholds (34 °C and 32 °C). In order to avoid unwarranted warnings, applying a region-specific threshold for the North is recommended. On the other hand, the East had fewer years of data in our analysis, with much less population density than other regions, resulting in irregular RaRR patterns compared to those of other sub-regions. Thus, applying the threshold for the entirety of Taiwan (32.5 °C) to East Taiwan is recommended, in order to avoid bias due to the small sample size. In summary, the best region-specific WBGT thresholds chosen were 34 °C for the North and 32.5 °C for the other three sub-regions.

## 4. Discussion

### 4.1. Heat-Warning Threshold Selection and Advantages of Our Method

We selected proper heat-warning thresholds according to the novel RaRRs based on the RRs of a refined GAM. In typical epidemiological studies, all RR values are conceptually compared with 1; the statistically significant RR values greater than 1 are numerically compared with each other among different categories, when scientists want to understand the relative health impacts among different categories. Such typical RR comparison may not work well in the heat-health relationships, as RR values of different sex, age, or sub-region categories at the same WBGT reference (30 °C, in our case) were different. In order to make comparisons among different categories with the same base, the RaRR is defined as the ratio of RR at a certain threshold candidate against their own baseline RR at the same WBGT reference (30 °C). In short, the advantage of using RaRR is that health risk enhancement comparisons can be made across different categories for different heat indicators with different lag periods. The following discussion focuses on the comparison of RaRR values, not RR, at the selected thresholds among different categories.

Whether WBGT or temperature is more suitable for heat-health warnings was evaluated using the RaRR. For the immediate heat-health outcomes (i.e., heat-related emergency visits), substantially higher RaRR for different sex, age and sub-regions (except Central Taiwan) were observed with WBGT than with temperature. Moreover, only minor differences in RaRR were observed for emergency and hospital visits among different sex and age groups with temperature compared to those with WBGT. Taken together, these results revealed that WBGT is a more sensitive heat-health indicator than temperature, especially for vulnerable populations. Recent studies have demonstrated that WBGT is a better heat-stress indicator for the general public, compared with the temperature or apparent temperature [7,29,33]. The Hong Kong government has begun using a modified WBGT as one of their heat indices [33]. Japan and Australia have also presented the WBGT as one of the heat indicators for the general public [34,35]. The HEAT-SHIELD platform in Europe also uses WBGT as the heat indicator with indoor and outdoor workers as the targeted population [17]. Thus, the WBGT should be considered a proper heat indicator for use in heat-health warning systems for the general public as well as for outdoor workers. Since the studied population in this work is the general public, the identified thresholds are for the general public. Nevertheless, the same methodology can be applied for occupational database to identify proper thresholds specifically for outdoor workers.

A warning system with the threshold identified using all-cause mortality aims at reducing overall mortality, while those with thresholds determined using emergency and hospital visits target at reducing morbidity related to human well-being and work productivity [36,37]. Compared with that between heat and mortality, the relationship of heat with morbidity has been less documented, as reviewed by Li et al. [37]; however, morbidity occurs before mortality. Moreover, our results showed that heat-related emergency visits had the highest RaRR and most prominent increasing trends, along with increasing WBGT thresholds at lags of 0–2, followed by hospital visits and all-cause mortality. The highest RaRR for the whole of Taiwan (1.44) was observed for heat-related emergency visits at a WBGT of 32.5 °C with lag 0. Hence, selecting heat-warning thresholds on the basis of heat-related emergency visits is recommended.

Various RaRR values were found in the different sub-regions in Taiwan, an island with geographically distinct regional features. Previous studies have also demonstrated the spatial variation of heat-related health impacts [37,38,39,40]. Our method successfully obtained appropriate threshold candidates for all sub-regions in Taiwan having different population sizes and areas, with the exception of East Taiwan. Compared with other sub-regions, East Taiwan has a smaller population and much lower population density (Table 1), with very few counts in the three health outcomes and only 13 years of data (2005–2017) available for modeling (Table 2). As a result, East Taiwan had few RRs with statistical significance and showed inconsistent trends of increase in RaRR. Thus, the threshold for Taiwan was suggested to be applied to East Taiwan in the heat warning system, to avoid small sample size bias. In short, it is important to identify area-specific thresholds, in order to effectively reduce heat-health risks in that area; however, caution should be taken when applying the present method in cities and countries with small sample sizes. The flexibility in selecting thresholds at city, regional, or country scales, whichever is more suitable, is another advantage of our method.

Thresholds for heat-health impacts in Taiwan have been assessed using different heat indicators, mostly temperature [29,41,42,43]. A recent study has also attempted to identify the temperature with the lowest health risks [44]. Again, these works have focused on determining the points of minimal health risks, rather than thresholds for a heat warning system. Such lowest identified health risks can serve as the reference in our method. Our previous work identified the heat-warning threshold of WBGT at 33 °C for the whole of Taiwan using risk comparison against the reference in terms of “difference,” without considering the lag effect and rare events [7]. We further refined the statistical model by evaluating risk comparisons with ratios (RaRRs), taking into consideration the lag effect of 0–2 days, and removing rare event interference. RaRR emphasizes the substantial enhancement of health risks. It is mathematically and conceptually superior than the “difference” comparison. Thus, the use of RaRR to identify thresholds for a heat warning system is recommended. Moreover, longer lag periods of heat-health relationship have been evaluated, up to 25 or 30 days; nevertheless, the most significant morbidity and mortality impacts have been found to occur in the first two days immediately after exposure to heat [37,40,44]. Thus, applying our method with RaRR considering lag effects of only two days was considered sufficient for selecting thresholds.

### 4.2. Consideration for Different Sex and Age Groups

As for assessing health impacts using WBGT on different sex groups, our analysis showed mixed results of either higher heat-health risks for males or females in morbidity or mortality, similar to those presented in the literature [40,45,46,47]. A review study has concluded that heat-related health impacts could be higher for males or females, in terms of different health outcomes [37]. Other socio-demographic factors may play a role, such as education level, marital status, and number of household occupants [46].

For heat and mortality/morbidity in different age groups, most studies have found the elderly to be the most vulnerable group, followed by children [11,36,37,38,39,46,48,49]. Our results found that, for emergency visits, children had the highest RaRR at lags of 0–1 (1.66, and 1.81, respectively) among the three age groups, while the elderly was the only group had statistically significant health impacts at lag 2. In other words, the elderly had longer lag effects in emergency visits. For hospital visits, children and the elderly had very similar RaRR at lags of 0–2. Additionally, for all-cause mortality, the elderly was the only group with significant RaRRs at both lags of 0 and 1 (1.03 and 1.05, respectively) among the three groups. Due consideration given to different health outcomes and lag days of these vulnerable populations is crucial when designing actions responsive to the heat-health warning, in order to reduce their risks.

Air conditioning may be used to reduce the heat-health risks faced by the elderly and children in the indoor environment. However, it is not a preferred choice due to the required energy usage. In urban areas, urban heat island effect would aggravate the heat accumulation in cities [50]. Proper urban planning may reduce health risks by reducing heat generation from urban infrastructures which leads to subsequent reduction of heat in both indoor and outdoor environments. Studies have shown that proper green-space planning and mixed land-use design may reduce cardiovascular mortality via reducing the effects of air pollution and heat [51,52]. Other health adaptation strategies or proper responsive actions may be developed without energy usage to protect vulnerable populations suffering from heat.

Furthermore, it should be noted that the RaRR with WBGT for heat-related hospital visits of those aged 15–64 (1.19) was higher than that for the elderly (1.16) and children (1.15) at lag 0. This age group was also the only group with significant RaRR (1.05) at lag 2 for all-cause mortality. The high RaRR in this age group—of mostly working adults—may be related to their higher chances of exposure to high heat under the sun while engaged in outdoor activities [37]. Formulating effective preventive measures to lower the risk of the working-age group are equally important as protecting the vulnerable groups. As most studies emphasized in health risks of vulnerable population, the health risks of high-exposure groups may be neglected by the authorities and the general public. The high RaRR of the 15–64 age group demonstrated in this work is an important indication of the lack of countermeasures targeting this seeming “not-vulnerable” but “high-exposure” group. Proper attention and actions are required to truly “leave no one behind” in reaching the SDGs.

### 4.3. Applicability and Limitation of Our Method

Attempts have been made to identify thresholds in different countries to facilitate heat-health intervention measures for the general public. For example, in the USA, Petitti et al. [20] attempted to determine multiple trigger points for activating heat-health intervention measures with multiple health outcomes. In Brisbane, Australia, Tong et al. [53] assessed the RR of daily mortality and emergency hospital admissions of previous heatwave events and proposed a tiered heat warning system according to health risks under a heatwave. In South Korea, Kang et al. [38] applied meta-analysis to define heat waves, using the results from a method considering lag-cumulative heatwave-related mortality risk at district levels. The current work provides another practical alternative, factoring in the considerations of policy makers.

After understanding the concerns of policy makers, we revised the statistical method with a novel RaRR as a systematic scientific tool for selecting a proper heat indicator (i.e., WBGT) with proper thresholds. This scientific tool takes into consideration substantial enhancement of health risks at lags of 0 to 2, removes rare-event interference, and accommodates at-risk populations of different sizes in different spatial scales. The occurring frequency of heat-alerts above the WBGT threshold with the highest RaRR was examined, in comparison against summer-long alerts, as it would be if based on minimal health risks identified by typical epidemiological studies. Our selected thresholds meet the criteria of both scientists and policy makers, having “statistically significant” and “substantial health impacts”. Thus, this proposed method can facilitate the establishment of a heat warning system in Taiwan, as well as in other countries. The low- and middle-income countries disproportionately affected by climate change [54] and located in tropical/sub-tropical areas are in urgent need of establishing such a heat warning system with their local health records, in particular.

There are two main limitations in this study. First, although Taiwan does not have an official heat warning system, the government has emphasized heat-protection measures to the public in the daily weather report if the forecasted temperature is high (particularly, above 37 °C) in the past 5–6 years. The data set analyzed covered 18 years; and the alerts issued in recent years may have changed people’s behaviors and reduced the number of recent cases in the health database. Second, adaptation measures taken on hot days in Taiwan were not taken into account. Taiwan has a 93% prevalence of air-conditioning [55]. Almost all office buildings during the daytime and most households in the evening turn on air-conditioning in summer. These adaptation measures may affect the analysis, giving statistically insignificant results even under a high heat-stress outdoor condition. Another minor limitation is the availability of mortality data which covered a shorter period. Despite these limitations, which would bias the results towards null, the present findings did show statistically significant RRs in most of the assessments, demonstrating significant heat-health impacts.

## 5. Conclusions

In response to the call from the UN for partnership for all to meet the SDG3 in reducing health risks due to rapid climate change, we presented a refined method with policy considerations in order to assist central and local governments worldwide in identifying proper heat-warning thresholds according to local health evidence. For demonstration, Taiwanese meteorological and health data over 18 years were analyzed. WBGT was recommended as the best for a heat warning system, although the proposed approach can be applied to any heat indicator. The WBGT has been used for six decades as an indicator for preventing heat stress-related health impacts in workplaces. New thresholds for WBGT in a heat-health warning system are needed in order to protect vulnerable populations in the general public, such as those aged 0–14 and ≥65 years. With the present method, proper thresholds for heat-warning systems can be identified so that excess health risks can be reduced with early warnings. The proposed approach can also facilitate other countries in establishing WBGT-based heat-health warning systems for the general public. This approach can be used by any country with available daily meteorological and health records. The flexibility of our method in selecting thresholds at city or regional scales is also demonstrated. However, caution should be taken when applying the method in cities and countries with small sample sizes. To interact with local authority which is responsible for establishing a heat warning system is highly recommended to streamline science-policy translation.

This refined approach assesses the heat-health relationships of emergency/hospital visits and mortality using a modified GAM with a lag effect of 0–2 days taken into account, compares the RR of different threshold candidates with that of a reference using RaRR, and expands statistical models by removing rare-event interference and providing the flexibility of application at different spatial scales. The novel RaRR allows for comparison of health risks across different categories, providing a solid scientific basis for threshold selection. The results showed that heat-related emergency visits are the most sensitive heat-related health record, and should thus be employed, if available, to select heat-warning thresholds. For the whole of Taiwan, the highest RaRR with WBGT occurred at a lag 0 for both heat-related emergency (1.44) and hospital visits (1.19) and at lag 1 for all-cause mortality (1.05). For different age groups, the highest RaRR occurred at different lag days with different health outcomes. Our results showed that children had the highest RaRR in emergency visits at lags of 0–1 (1.66 and 1.81, respectively) and in hospital visits at lag 1 (1.25), while the elderly was the only age group that had significantly high RaRR in emergency visits at lags 0–2 (1.25–1.48) and in all-cause mortality at lags 0–1 (1.03–1.05). It should be emphasized that those aged 15–64 had the highest RaRR in heat-related hospital visits at lag 0; and this age group was the only group with significant RaRR (1.05) at lag 2 for all-cause mortality. Proper attention and actions are required for different “vulnerable” or “high-exposure” sub-populations, in order to truly “leave no one behind” in reaching the SDGs. Moreover, in view of the spatial variations of heat-health impacts, threshold identification should be conducted for specific cities/regions. Based on the RaRR of heat-related emergency visits, the recommended WBGT thresholds were 34 °C for North Taiwan and 32.5 °C for the other three sub-regions. In summary, this work presents a systematic scientific tool for policy makers to select a proper threshold to facilitate the implementation of an effective heat warning system for better heat-stress adaptation, in order to reduce health risks under climate change.

The SDG3 is to ensure healthy lives and promote well-being for all at all ages. An evidence-based heat-health warning system established based on thresholds identified by our approach is certain an effective way to reduce heat-health related risks for all of all ages for reaching SDG3. By co-creating knowledge with stakeholders, we revised statistical models to provide practical thresholds which can be readily used in establishing a heat warning system. In other words, translation from scientific evidence to policies were facilitated with stakeholder engagement throughout the research process. There are other health adaptation strategies which can reduce climate-related health risks such as urban planning requiring in-depth stakeholder engagement to ensure smooth science-policy translation. The current work demonstrated a successful example of co-creating knowledge with stakeholders. Hopefully, there will be more successful cases in the future to reduce climate-related health risks for all at all ages.

## Figures and Tables

**Figure 1 ijerph-18-09506-f001:**
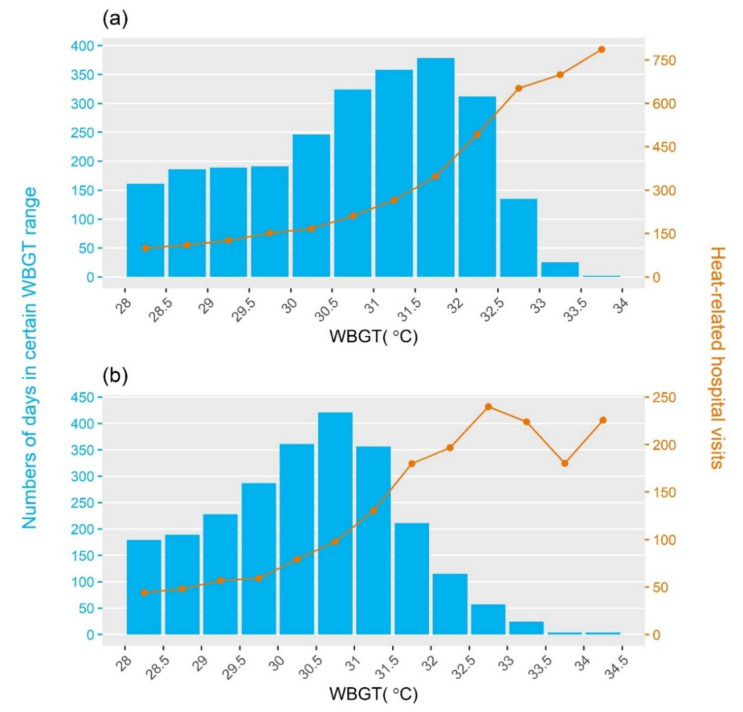
Relationship of WBGT with heat-related hospital visits for (**a**) whole of Taiwan and (**b**) Central Taiwan. The blue bars denote numbers of days in a certain WBGT range, and the orange line indicates numbers of heat-related hospital visits. Rare events of WBGT above 33 °C in Central Taiwan may cause fluctuations in counts of heat-related hospital visits.

**Figure 2 ijerph-18-09506-f002:**
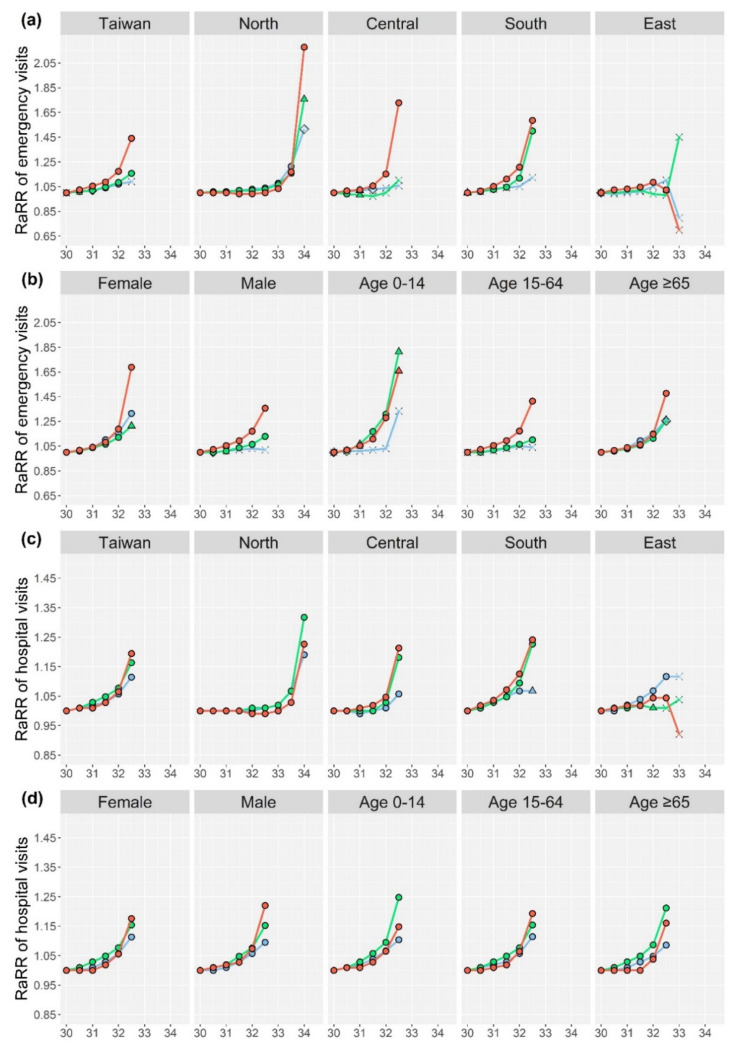
Reference-adjusted risk ratios (RaRRs) of heat-related emergency visits, heat-related hospital visits, and all-cause mortality at different WBGT threshold candidates for the whole of Taiwan and different sub-regions (**a**,**c**,**e**), and different sex and age (**b**,**d**,**f**). *p*-values are those for the corresponding relative risks at that threshold candidate.

**Figure 3 ijerph-18-09506-f003:**
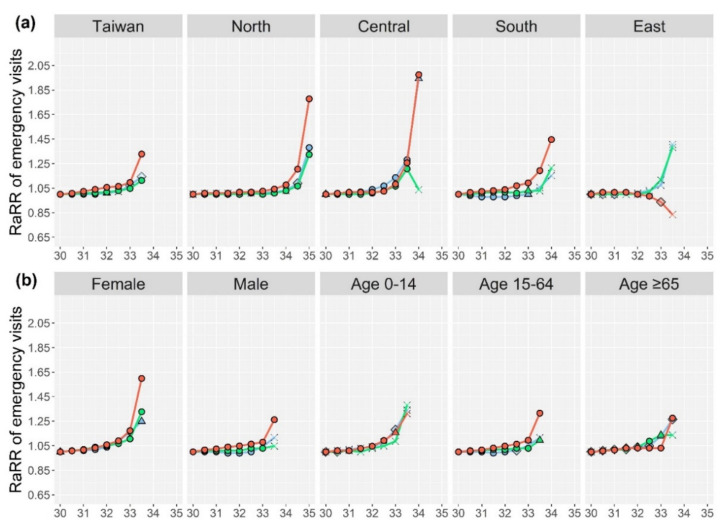
Reference-adjusted risk ratios (RaRRs) of heat-related emergency visits, heat-related hospital visits, and all-cause mortality at different temperature threshold candidates for the whole of Taiwan and different sub-regions (**a**,**c**,**e**), and different sex and age (**b**,**d**,**f**). *p*-values are those for the corresponding relative risks at that threshold candidate.

**Figure 4 ijerph-18-09506-f004:**
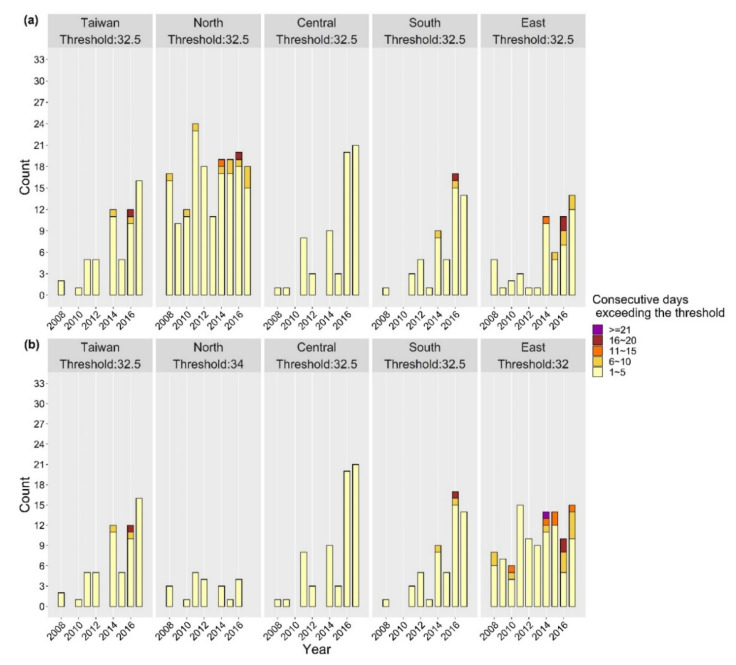
Counts of periods of consecutive days exceeding the specified WBGT thresholds in different sub-regions of Taiwan between 2008 and 2017; (**a**) the same threshold for all sub-regions and (**b**) region-specific threshold.

**Table 1 ijerph-18-09506-t001:** Characteristics of Taiwan and sub-regions with 99.5th percentile of WBGT and temperature and chosen limits.

Variable		Taiwan	North	Central	South	East
Population		23,316,818	10,776,529	4,570,579	6,964,326	1,005,384
Area (km^2^)		355,887	7030	7396	11,174	10,287
Population density (Pop. per km^2^)		650	1533	618	623	98
WBGT	99.5th percentile (2000–2017)	33.1	34.3	33.0	33.2	33.4
	99.5th percentile (2008–2017) ^a^	33.2	34.3	33.5	33.2	33.4
	Chosen upper limit	33.0	34.5	33.0	33.0	33.5
Temperature	99.5th percentile (2000–2017)	34.1	35.5	34.3	34.3	34.0
	99.5th percentile (2008–2017) ^a^	34.2	35.6	34.4	34.3	33.9
	Chosen upper limit	34.0	35.5	34.5	34.5	34.0

^a^ 2008–2017 is the duration analyzed for all-cause mortality.

**Table 2 ijerph-18-09506-t002:** Summary of warm-season (a) health and (b) environmental data analyzed (May to October 2000–2017).

**(a)** **Category**	**Daily Emergency Visits** **(*n* ^b^)**				**Daily Hospital Visits** **(*n* ^b^)**				**Daily All-Cause Mortality ^a^** **(*n* = 1840)**			
	**Mean**	**SD**	**Min**	**Max**	**Mean**	**SD**	**Min**	**Max**	**Mean**	**SD**	**Min**	**Max**
Whole of Taiwan	8	9	0	73	222	266	1	1533	357	56	187	510
Sex												
Female	2	3	0	26	126	155	0	855	145	26	67	230
Male	6	6	0	45	93	109	0	673	212	33	113	291
Age (years)												
0–14	1	1	0	7	9	9	0	63	3	2	0	10
15–64	6	7	0	51	189	231	0	1351	94	16	42	140
≥65	1	2	0	19	23	28	0	176	260	44	131	390
Sub-region												
North Taiwan	3	4	0	41	117	149	0	917	148	19	93	218
Central Taiwan	2	2	0	24	81	94	0	510	67	16	15	113
South Taiwan	2	3	0	23	17	20	0	152	121	30	25	182
East Taiwan ^c^	1	2	0	15	9	11	0	67	21	5	7	40
**(b)** **Category**	**Daily Maximum WBGT (°C)** **(*n* ^b^)**				**Daily Maximum Temperature (°C)** **(*n* ^b^)**				**Daily Mean PM_2.5_ (μg/m^3^)** **(*n* ^d^)**			
	**Mean**	**SD**	**Min**	**Max**	**Mean**	**SD**	**Min**	**Max**	**Mean**	**SD**	**Min**	**Max**
Whole Taiwan island	29.6	2.4	19.5	33.6	30.7	2.2	21.0	34.7	25.4	11.1	5.9	92.0
North Taiwan	29.4	3.3	16.3	35.5	30.6	3.2	18.3	37.1	22.1	10.7	2.1	106.5
Central Taiwan	29.3	2.2	19.2	34.3	30.7	2.1	19.9	37.9	27.8	14.9	5.0	139.7
South Taiwan	30.0	1.9	21.1	34.0	31.3	1.8	22.2	34.9	28.1	15.6	5.9	207.1
East Taiwan ^c^	29.7	2.6	19.0	33.8	30.3	2.2	21.4	34.7	14.7	7.6	4.3	72.7

^a^: Analysis for all-cause mortality was for 2008–2017 only, due to locations of death not being specified for 2000–2007. ^b^: *n* = 3312 for North, Central, and South Taiwan; *n* = 2392 for East Taiwan. ^c^: Analysis for East Taiwan was for 2005–2017 only, due to no PM_2.5_ measurements for 2000–2004. ^d^: *n* = 3312, 3207, 3275, 3243, and 2392 for the whole of Taiwan, North, Central, South, and East Taiwan, respectively. SD: standard deviation; Min: minimum; Max: Maximum.

**Table 3 ijerph-18-09506-t003:** Relative risks (RRs) at different (a) WBGT and (b) temperature threshold candidates for the whole of Taiwan.

**(a) WBGT**	**RR (95%CI) ^a^**					
**Threshold** **(°C)**	**30**	**31**	**31.5**	**32**	**32.5**	
**Heat-related emergency visits**						
Lag 0	1.27 *** (1.26,1.29)	1.34 *** (1.32,1.36)	1.38 *** (1.35,1.42)	1.49 *** (1.44,1.54)	**1.83 *** (1.68,1.99)**	
Lag 1	1.08 *** (1.07,1.10)	1.10 *** (1.08,1.13)	1.13 *** (1.10,1.16)	1.17 *** (1.13,1.22)	**1.25 *** (1.14,1.37)**	
Lag 2	1.00 (0.99,1.01)	1.02 * (1.00,1.04)	1.04 *** (1.02,1.07)	1.07 *** (1.03,1.11)	**1.09 (0.99,1.20)**	
**Heat-related hospital visits**						
Lag 0	1.08 *** (1.08,1.09)	1.09 *** (1.09,1.10)	1.11 *** (1.10,1.11)	1.15 *** (1.14,1.16)	**1.29 *** (1.27,1.31)**	
Lag 1	1.04 *** (1.04,1.05)	1.07 *** (1.07,1.07)	1.09 *** (1.09,1.10)	1.12 *** (1.12,1.13)	**1.21 *** (1.19,1.23)**	
Lag 2	1.05 *** (1.05,1.05)	1.07 *** (1.06,1.07)	1.08 *** (1.08,1.09)	1.11 *** (1.11,1.12)	**1.17 *** (1.15,1.19)**	
**All-cause mortality**						
Lag 0	1.00 *** (1.00,1.01)	1.00 * (1.00,1.01)	1.01 * (1.00,1.01)	1.01 ** (1.00,1.02)	**1.03 ** (1.01,1.05)**	
Lag 1	1.00 ** (1.00,1.01)	1.01 *** (1.00,1.01)	1.01 *** (1.01,1.02)	1.03 *** (1.02,1.04)	**1.05 *** (1.03,1.08)**	
Lag 2	1.00 (1.00,1.00)	1.00 * (1.00,1.01)	1.01 * (1.00,1.01)	1.01 (1.00,1.02)	**1.03 * (1.00,1.05)**	
**(b) Temperature**	**RR (95%CI) ^a^**					
**Threshold** **(°C)**	**30**	**31**	**32**	**32.5**	**33**	**33.5**
**Heat-related emergency visits**						
Lag 0	1.25 *** (1.24,1.26)	1.28 *** (1.26,1.29)	1.32 *** (1.29,1.34)	1.33 *** (1.30,1.36)	1.37 *** (1.33,1.42)	**1.66 *** (1.52,1.81)**
Lag 1	1.06 *** (1.04,1.07)	1.07 *** (1.05,1.08)	1.08 *** (1.06,1.10)	1.09 *** (1.07,1.12)	1.11 *** (1.06,1.15)	**1.18 *** (1.07,1.29)**
Lag 2	0.97 *** (0.96,0.98)	0.97 *** (0.96,0.98)	0.98 ** (0.96,0.99)	0.99 (0.96,1.01)	1.04 * (1.00,1.08)	**1.11 * (1.01,1.22)**
**Heat-related hospital visits**						
Lag 0	1.10 *** (1.10,1.10)	1.11 *** (1.10,1.11)	1.12 *** (1.12,1.12)	1.14 *** (1.13,1.14)	1.18 *** (1.17,1.19)	**1.31 *** (1.29,1.33)**
Lag 1	1.03 *** (1.03,1.03)	1.04 *** (1.04,1.04)	1.06 *** (1.06,1.07)	1.08 *** (1.08,1.09)	1.10 *** (1.09,1.11)	**1.12 *** (1.10,1.14)**
Lag 2	1.04 *** (1.04,1.04)	1.04 *** (1.04,1.04)	1.04 *** (1.04,1.04)	1.05 *** (1.04,1.05)	1.07 *** (1.06,1.08)	**1.15 *** (1.13,1.18)**
**All-cause mortality**						
Lag 0	1.00 *** (1.00,1.01)	1.00 *** (1.00,1.01)	1.01 *** (1.00,1.01)	1.01 *** (1.01,1.02)	1.02 *** (1.01,1.03)	**1.05 *** (1.02,1.07)**
Lag 1	1.00 (1.00,1.00)	1.00 * (1.00,1.01)	1.01 ** (1.00,1.01)	1.01 * (1.00,1.02)	1.01 * (1.00,1.02)	**1.04 ** (1.01,1.07)**
Lag 2	1.00 * (1.00,1.00)	1.00 * (1.00,1.01)	1.01 *** (1.00,1.01)	1.01 *** (1.01,1.02)	1.02 *** (1.01,1.03)	**1.04 ** (1.02,1.07)**

^a^: Relative risk (RR) and 95% confidence interval (95% CI) of each outcome with a 0.5 °C increase above the threshold. *, *p* < 0.05; **, *p* < 0.01; ***, *p* < 0.001.

## Data Availability

The health records contain sensitive individual-level information, which is not publicly available. It can be made available to researchers after approval of a formal application to the Health and Welfare Data Science Center of the Ministry of Health and Welfare. The meteorological and air pollutant data can be obtained with a formal application to the Taiwan Central Weather Bureau and Taiwan Environmental Protection Administration, respectively.

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
