# Peer review of "Selecting Thresholds of Heat-Warning Systems with Substantial Enhancement of Essential Population Health Outcomes for Facilitating Implementation"

_ijerph, 2021, doi:10.3390/ijerph18189506_

Round 1

Reviewer 1 Report

A well-written and interesting paper. I have made some suggestions that you will find in the attached document. 

Reviewer 2 Report

This is an interesting paper which refines a generalized additive model for selecting thresholds with substantial health risk enhancement, based on 18-year records of a population database, considering lag effects and different spatial scales. Reference-adjusted risk ratio (RaRR) is proposed, defined as the ratio between the relative risk of an essential health outcome for a threshold candidate against that for a reference; the threshold with the highest RaRR is potentially the optimal one.  Despite the noted strengthens, just a couple of minor observations that would indeed enhance the quality of the revised manuscript.

Abstract: This should be restructured to tell the story around the following key areas: Background, aim, methods and materials or methodology, results, conclusions. Some indication of the actual time frame or period for the “18-year records of a population” would enhance the quality of the abstract. Further, some of the information included within the ‘conclusion’ section such as the following first sentence “In response to the call from the UN for partnership for all to meet the SDG3 in reducing health risks due to rapid climate change” could also be used to strengthen the background aspect of the abstract.

Justification for the selection of the records- The authors should include detailed justification for the selection of the records and associated timeline of between 2000 and 2017 as obtained from the database of the Health and Welfare Data Science Center of the Ministry of Health and Welfare. Further, some detailed explanations of why the all-cause mortality counts (excluding accidents and suicide) needed to start 2008 and 2017 as obtained from the Taiwan National Mortality Registry. Data?

Conclusions: The generalisation of the proposed and refined statistical method for identifying proper thresholds should be commented upon. Further, some of the identified implications’ as listed within the introduction section should be extended to and elaborated upon within the conclusion sections.  Finally, the conclusion section needs to be revisited through inclusion of the emergent contributions to knowledge as well as the limitations of the study should be reported. Currently, this section is very narrow.
